# Learning the Pareto Front with Hypernetworks

**Aviv Navon***
Bar-Ilan University, Israel
`aviv.navon@biu.ac.il`

**Aviv Shamsian***
Bar-Ilan University, Israel
`aviv.shamsian@biu.ac.il`

**Ethan Fetaya**[†]
Bar-Ilan University, Israel
`ethan.fetaya@biu.ac.il`

**Gal Chechik**[†]
Bar-Ilan University, Israel
NVIDIA, Israel
`gal.chechik@biu.ac.il`

## Abstract

Multi-objective optimization (MOO) problems are prevalent in machine learning. These problems have a set of optimal solutions, called the Pareto front, where each point on the front represents a different trade-off between possibly conflicting objectives. Recent MOO methods can target a specific desired ray in loss space however, most approaches still face two grave limitations: (i) A separate model has to be trained for each point on the front; and (ii) The exact trade-off must be known before the optimization process. Here, we tackle the problem of learning the entire Pareto front, with the capability of selecting a desired operating point on the front after training. We call this new setup *Pareto-Front Learning* (PFL).

We describe an approach to PFL implemented using HyperNetworks, which we term *Pareto HyperNetworks* (PHNs). PHN learns the entire Pareto front simultaneously using a single hypernetwork, which receives as input a desired preference vector and returns a Pareto-optimal model whose loss vector is in the desired ray. The unified model is *runtime efficient* compared to training multiple models and generalizes to new operating points not used during training. We evaluate our method on a wide set of problems, from multi-task regression and classification to fairness. PHNs learn the entire Pareto front at roughly the same time as learning a single point on the front and at the same time reach a better solution set. PFL opens the door to new applications where models are selected based on preferences that are only available at run time.

## 1 Introduction

Multi-objective optimization (MOO) aims to optimize several possibly conflicting objectives. MOO is abundant in machine learning problems, from multi-task learning (MTL), where the goal is to learn several tasks simultaneously, to constrained problems. In such problems, one aims to learn a single task while finding solutions that satisfy properties like fairness or privacy. It is common to optimize the main task while adding loss terms to encourage the learned model to obtain these properties.

MOO problems have a set of optimal solutions, the *Pareto front*, each reflecting a different trade-off between objectives. Points on the Pareto front can be viewed as an intersection of the front with a specific direction in loss space (a ray, Figure 1). We refer to this direction as a *preference vector*, as it represents a single trade-off between objectives. When a direction is known in advance, it is possible to obtain the corresponding solution on the front (Mahapatra & Rajan, 2020).

However, in many cases, we are interested in more than one predefined direction, either because the trade-off is not known before training, or because there are many possible trade-offs of interest. For

---

*Equal contributor
[†]Equal contributor

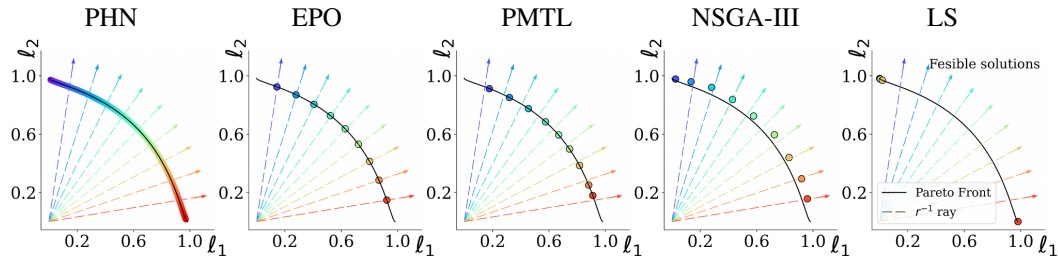

Figure 1: *Illustrative example using the popular task of Fonseca (1995)*: demonstrating the relation between Pareto front, preference rays, and solutions. Pareto front (black solid line) for a 2D loss space and several rays (colored dashed lines) which represent various possible preferences. **Left:** A single PHN-EPO model learns the entire Pareto front, mapping any given preference ray to its corresponding solution on the front. The data and task are detailed in Section 5.1.

example, network routing optimization aims to maximize bandwidth, minimize latency, and obey fairness. However, the cost and trade-off vary from one application running in the network to another, or even continuously change in time. The challenge remains to design a model that can be applied at inference time to any given preference direction, even ones not seen during training. We call this problem *Pareto front learning* (PFL).

Although several recent studies (Yang et al., 2019; Parisi et al., 2016) suggested to learn the trade-off curve in MOO problems, there is no existing scalable and general-purpose MOO approach that provides Pareto-optimal solutions for numerous preferences in objective space. Classical approaches, like genetic algorithms, do not scale to modern high-dimensional problems. It is possible in principle to run a single-direction optimization multiple times, each for a different preference, but this approach faces two major drawbacks: (i) *Scalability* – the number of models to be trained to cover the objective space grows exponentially with the number of objectives; and (ii) *Flexibility* – the decision maker cannot switch freely between preferences unless all models are trained and stored in advance.

Here we put forward a new view of PFL as a problem of learning a conditional model, where the conditioning is over the preference direction. During training, a single unified model is trained to produce Pareto optimal solutions while satisfying the given preferences. During inference, the model covers the Pareto front by varying the input preference vector. We further describe an architecture that implements this idea using HyperNetworks. Specifically, we train a hypernetwork, termed *Pareto Hypernetwork* (PHN), that given a preference vector as an input, produces a deep network model tuned for that objective preference. Training is applied to preferences sampled from the $m-$dimensional simplex where $m$ represents the number of objectives.

We evaluate PHN on a wide set of problems, from multi-class classification, through fairness and image segmentation to multi-task regression. We find that PHN can achieve superior overall solutions while being $10 \sim 50$ times faster (see Figure 5). PHN addresses both scalability and flexibility. Training a unified model allows using any objective preference at inference time. Finally, as PHN generates a continuous parametrization of the entire Pareto front, it could open new possibilities to analyze Pareto optimal solutions in large-scale neural networks.

Our paper has the following contributions: **(1)** We define the *Pareto-front learning* problem – learn a model that at inference time can operate on any given preference vector, providing a Pareto-optimal solution for that specified objective trade-off. **(2)** We describe *Pareto Hypernetworks* (PHN), a unified architecture based on hypernetworks that addresses PFL and shows it can be effectively trained. **(3)** Empirical evaluations on various tasks and datasets demonstrate the ability of PHNs to generate better objective space coverage compared to multiple baseline models, with significant improvement in training time.

## 2 MULTI-OBJECTIVE OPTIMIZATION

We start by formally defining the MOO problem. An MOO is defined by $m$ losses $\ell_i : \mathbb{R}^d \to \mathbb{R}_+$, $i = 1, \ldots, m$, or in vector form $\boldsymbol{\ell} : \mathbb{R}^d \to \mathbb{R}_+^m$. We define a partial ordering on the loss space

$\mathbb{R}_+^m$ by setting $\boldsymbol{\ell}(\theta_1) \preceq \boldsymbol{\ell}(\theta_2)$ if for all $i \in [m]$, $\ell_i(\theta_1) \leq \ell_i(\theta_2)$. We define $\boldsymbol{\ell}(\theta_1) \prec \boldsymbol{\ell}(\theta_2)$ if $\boldsymbol{\ell}(\theta_1) \preceq \boldsymbol{\ell}(\theta_2)$ and for some $i \in [m], \ell_i(\theta_1) < \ell_i(\theta_2)$. We say that a point $\theta_1 \in \mathbb{R}^d$ *dominates* $\theta_2 \in \mathbb{R}^d$ if $\boldsymbol{\ell}(\theta_1) \prec \boldsymbol{\ell}(\theta_2)$.

If a point $\theta_1$ dominates $\theta_2$, then $\theta_1$ is clearly preferable, because it improves some objectives and is not worse on any other objective. Otherwise, solutions present a certain trade-off and selecting one specific solution requires additional information about the user preferences. A point that is not dominated by any other point is called *Pareto optimal*. The set of all Pareto optimal points is called the *Pareto front*. Since many modern machine learning models, e.g., deep neural networks, use non-convex optimization, one cannot expect global optimality. We call a point *local Pareto optimal* if it is Pareto optimal in some open neighborhood of the point.

The most straightforward approach to MOO is linear scalarization (LS), where one defines a new single loss $\ell_{\boldsymbol{r}}(\theta) = \sum_i r_i \ell_i(\theta)$ given a vector $\boldsymbol{r} \in \mathbb{R}_+^m$ of weights. One can then apply standard, single-objective optimization algorithms. LS has two major limitations. First, it can only reach the convex part of the Pareto front (Boyd et al., 2004, Chapter 4.7), as shown empirically in Figure 1. Second, if one wishes to target a specific ray in loss space, specified by a preference vector, it is not clear which linear weights lead to that desired Pareto optimal point.

In the context of the current work, we highlight two properties that an MOO optimization procedure should possess: scale to many objectives and control which solution on the Pareto front is obtained. Lin et al. (2019) described *Pareto multi-task learning* (PMTL), an algorithm that splits the loss space into separate cones based on the selected reference rays, and returns a solution per cone using a constrained version of Fliege & Svaiter (2000). This approach allows the user to target several points on the Pareto front; However, it scales poorly with the number of cones and does not converge to the exact desired ray on the Pareto front.

Convergence to the desired ray in loss space can be achieved using *Exact Pareto Optimal* (EPO) (Mahapatra & Rajan, 2020). To find the intersection of the Pareto front with a given preference ray $\boldsymbol{r}$, EPO balances two goals: Finding a descent direction towards the Pareto front and approaching the desired ray. EPO searches for a point in the convex hull of the gradients, known by Désidéri (2012) to include descent directions, that has a maximal angle with a vector $d_{bal}$ which pulls the point to the desired ray. EPO combines gradient descent and controlled ascent enabling it to reach an exact Pareto optimal solution if one exists, or the closest Pareto optimal solution.

## 3  RELATWED WORK

**Multitask learning.**   In multitask learning (MTL) we simultaneously solve several learning problems while sharing information among tasks (Zhang & Yang, 2017; Ruder, 2017). In some cases, MTL-based models outperform their single task counterparts in terms of per-task performance and computational efficiency (Standley et al., 2019). MTL approaches map the loss vector into a single loss term using a fixed or dynamic weighting scheme. The most frequently used approach is Linear Scalarization (LS), in which each loss term's weight is chosen apriori. A proper set of weights is commonly selected using grid search. Unfortunately, such an approach scales poorly with the number of tasks. Recently, MTL methods propose dynamically balance the loss terms using gradient magnitude (Chen et al., 2018), the rate of change in losses (Liu et al., 2019), task uncertainty (Kendall et al., 2018), or learning non-linear loss combinations by implicit differentiation (Navon et al., 2020). However, those methods seek a balanced solution and are not suitable for modeling task trade-offs.

**Multi-objective optimization.**   The goal of Multi-Objective Optimization (MOO) is to find Pareto optimal solutions corresponding to different trade-offs between objectives (Ehrgott, 2005). MOO has a wide variety of applications in machine learning, spanning Reinforcement Learning (Van Moffaert & Nowé, 2014; Pirotta & Restelli, 2016; Parisi et al., 2014; Pirotta et al., 2015; Parisi et al., 2016; Yang et al., 2019), neural architecture search (Lu et al., 2019; Hsu et al., 2018), fairness (Martínez et al., 2020; Liu & Vicente, 2020), and Bayesian optimization (Shah & Ghahramani, 2016; Hernández-Lobato et al., 2016). Genetic algorithms (GA) are a popular approach for small-scale multi-objective optimization problems. GAs are designed to maintain a set of solutions during optimization. Therefore, they can be extended in natural ways to maintain solutions for different rays. In this area, leading approaches include NSGA-III (Deb & Jain, 2013) and MOEA/D (Zhang & Li,

2007). However, these gradient-free methods scale poorly with the number of parameters and are not suitable for training large-scale neural networks.

Sener & Koltun (2018) proposed a gradient-based MOO algorithm for MTL, based on MDGA (Désidéri, 2012), suitable for training large-scale neural networks. Other recent works include (Lin et al., 2019; Mahapatra & Rajan, 2020) detailed in Section 2. Another recent approach that aims at a more complete view of the Pareto front is Ma et al. (2020). They extend a given Pareto optimal solution in its local neighborhood. In a concurrent work, Lin et al. (2020) extends Lin et al. (2019) for approximating the entire Pareto front. They train a single hypernetwork by constantly changing the reference directions in the PMTL objective. The proposed method is conceptually similar to our approach, but since it builds on PMTL, it may not produce an exact mapping between the preference and the corresponding solution. Similarly, Dosovitskiy & Djolonga (2019) proposed learning a single model conditioning on the objective weight vector. The method uses feature-wise linear modulation Perez et al. (2017), and dynamically weighted LS loss criterion.

**Hypernetworks.** Ha et al. (2017) introduced the idea of hypernetworks (HNs) inspired by the genotype-phenotype relation in cellular biology. HN presents an approach of using one network (hypernetwork) to generate weights for a second network (target network). In recent years, HN are widely used in various domains such as computer vision (Klocek et al., 2019; Ha et al., 2017), language modeling (Suarez, 2017), sequence decoding (Nachmani & Wolf, 2019), continual learning (Oswald et al., 2020), federated learning (Shamsian et al., 2021) and hyperparameter optimization (Mackay et al., 2019; Lorraine & Duvenaud, 2018). HN dynamically generates models conditioned on a given input, obtaining a set of customized models using a single learnable network.

## 4 PARETO HYPERNETWORKS

**Algorithm 1** PHN

> **while** not converged **do**
> $\quad \boldsymbol{r} \sim Dir(\boldsymbol{\alpha})$
> $\quad \theta(\phi, \boldsymbol{r}) = h(\boldsymbol{r}; \phi)$
> $\quad$ Sample mini-batch $(x_1, y_1), .., (x_B, y_B)$
> $\quad$ **if** LS **then**
> $\quad\quad g_\phi \leftarrow \frac{1}{B} \sum_{i,j} r_i \nabla_\phi \ell_i(x_j, y_j, \theta(\phi, \boldsymbol{r}))$
> $\quad$ **if** EPO **then**
> $\quad\quad \beta = EPO(\theta(\phi, \boldsymbol{r}), \boldsymbol{\ell}, \boldsymbol{r})$
> $\quad\quad g_\phi \leftarrow \frac{1}{B} \sum_{i,j} \beta_i \nabla_\phi \ell_i(x_j, y_j, \theta(\phi, \boldsymbol{r}))$
> $\quad \phi \leftarrow \phi - \eta g_\phi$
> **return** $\phi$

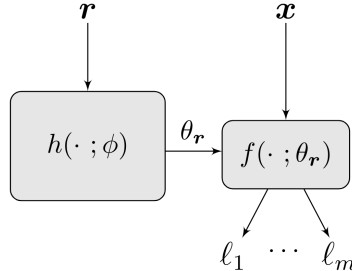

Figure 2: A PHN $h(\cdot; \phi)$ receives an input preference ray $\boldsymbol{r}$ and outputs the corresponding model weights $\theta_{\boldsymbol{r}}$.

We start by describing the basic of hypernetworks architecture (Figure 2). Hypernetworks are deep models that generate the weights of another deep model as their output, named the target network. The weights produced by the hypernetwork depend on its input, and one can view the training of a hypernetwork as training a family of target networks simultaneously, conditioned on the input. Let $h(\cdot; \phi)$ denote the hypernetwork with parameters $\phi \in \mathbb{R}^n$, and let $f$ denote the target network whose parameters we denote by $\theta$. In our implementation, we parametrize $h$ using a feed-forward network with multiple heads. Each head outputs a different weight tensor of the target network. More precisely, the input is first mapped to a higher dimensional space using an MLP, to construct shared features. These features are passed through fully connected layers to create a weight matrix per layer in the target network.

Consider an $m-$dimensional MOO problem with an objective vector $\boldsymbol{\ell}$. Let $\boldsymbol{r} = (r_1, ..., r_m) \in S_m$ denote a preference vector representing a desired trade-off over the objectives. Here $S_m = \{\boldsymbol{r} \in \mathbb{R}^m \mid \sum_j r_j = 1, r_j \geq 0\}$ is the $m-$dimensional simplex. Our PHN model takes as input the preference vector $\boldsymbol{r}$ to produce the weights $\theta_{\boldsymbol{r}} = \theta(\phi, \boldsymbol{r}) = h(\boldsymbol{r}; \phi)$ of the target network. The target network is then applied to the training data in a usual manner. Throughout the training, we randomly sample $\boldsymbol{r}$ at each iteration from a Dirichlet distribution with parameter $\alpha \in \mathbb{R}^m$. At inference time, a given preference direction $\boldsymbol{r}$, is provided as input to the HN $h$, which outputs the corresponding Pareto

optimal solution, $\theta_{\boldsymbol{r}}$, namely, a set of model weights for specific given $\boldsymbol{r}$. We train $\theta$ to be Pareto optimal and corresponds to the preference vector. For that purpose, we explore two alternatives.

**PHN-LS:** The first, named PHN-LS, uses *linear scalarization* with the preference vector $\boldsymbol{r}$ as loss weights, i.e the loss for input $\boldsymbol{r}$ is $\sum_i r_i \ell_i$. While linear scalarization has some theoretical limitations, it is a straightforward and fast approach that tends to work well in practice. One can think of the PHN-LS approach as optimizing the loss $\ell_{LS}(\phi) = \mathbb{E}_{\boldsymbol{r},x,y}[\sum_i r_i \ell_i(x, y, \theta(\phi, \boldsymbol{r}))]$ where we sample a mini-batch and a weighting $\boldsymbol{r}$. We note, as a direct result of the implicit function theorem, the following:

**Proposition 1.** *Let $\theta(\hat{\boldsymbol{r}})$ be a local Pareto optimal point corresponding to weights $\hat{\boldsymbol{r}}$, i.e. $\sum_{ij} \hat{r}_i \nabla \ell_i(x_j, y_j, \theta) = 0$ and assume the Hessian of $\sum_i \hat{r}_i \ell_i$ has full rank at $\theta(\hat{\boldsymbol{r}})$. There exists a neighborhood $U$ of $\hat{\boldsymbol{r}}$ and a smooth mapping $\theta(\boldsymbol{r})$ such that $\forall \boldsymbol{r} \in U : \sum_{ij} r_i \nabla \ell_i(x_j, y_j, \theta(\boldsymbol{r})) = 0$.*

This shows that under the full rank assumption the connection between the LS term $\boldsymbol{r}$ and the optimal model weights $\theta(\boldsymbol{r})$ is smooth and can be learned using universal approximation (Cybenko, 1989). We also observe that if for all $\boldsymbol{r}$, $\theta(\phi, \boldsymbol{r})$ is a local Pareto optimal point corresponding to $\boldsymbol{r}$, i.e., $\sum_{ij} r_i \nabla \ell_i(x_j, y_j, \theta(\boldsymbol{r})) = 0$, then $\nabla_\phi \ell_{LS}(\phi) = 0$ and this is a stationary point for $\ell_{LS}$. However, there could exists stationary points and even local minima such that $\nabla_\phi \ell_{LS}(\phi) = 0$ but in general $\sum_{ij} r_i \nabla \ell_i(x_j, y_j, \theta(\boldsymbol{r})) \neq 0$ and the output of our hypernetwork is not a local Pareto optimal point. While these bad local minima exist, as in standard deep neural networks, we empirically find that PHNs avoid these solutions and achieve good results.

**PHN-EPO:** Our second approach treats the preference $\boldsymbol{r}$ as a ray in loss space and trains $\theta(\phi, \boldsymbol{r})$ to reach a Pareto optimal point on the inverse ray $\boldsymbol{r}^{-1}$, namely, $r_1 \cdot \ell_1 = \ldots = r_m \cdot \ell_m$. This is achieved by using the EPO update direction for each output of the Pareto hypernetwork. The recent EPO algorithm can guarantee convergence to a local Pareto optimal point on a specified ray $\boldsymbol{r}$, addressing the theoretical limitations of LS. Since the EPO descent direction is a convex combination of the gradients, we use the EPO algorithm to compute this adaptive weighting and backpropagate the reweighted losses. See pseudo-code in Alg. 1.

The EPO however, unlike LS, cannot be seen as optimizing a certain objective. Therefore, we cannot consider the PHN-EPO optimization, which uses the EPO descent direction, as optimizing a specific objective. We also note that since the EPO needs to solve a linear programming step at each iteration, the runtime of EPO and PHN-EPO are longer than the LS and PHN-LS counterparts (see Figure 5).

**PHN Scalability:** Using a naive implementation of HNs, their size grows linearly with the target network's size; this may restrict the scalability of PHNs to large target networks. Fortunately, our approach can be easily extended to handle large target networks. First, one can learn only part of the target network parameters using the HN (Luan et al., 2018), or learn preference-specific normalization layers (Huang & Belongie, 2017; Perez et al., 2017) as in Dosovitskiy & Djolonga (2019). Furthermore, one can apply "chunking": generate different parts of the target network, conditioned on a trainable chunk descriptor input (Oswald et al., 2020). Similarly, one can reduce the dimension of the shared preference representation to achieve a similar effect (see Appendix B.5). The experiments in the main paper used the vanilla version of PHN.

## 5 EXPERIMENTS

We evaluate Pareto HyperNetworks (PHNs) on a set of diverse multi-objective problems. The experiments show the superiority of PHN over previous MOO methods. We make our source code publicly available at: https://github.com/AvivNavon/pareto-hypernetworks.

**Compared methods:** We compare the following approaches: **(1)** Our proposed Pareto HyperNetworks (**PHN-LS** and **PHN-EPO**) algorithm described in Section 4; **(2)** Linear scalarization (**LS**); **(3)** Pareto MTL (**PMTL**) Lin et al. (2019); **(4)** Exact Pareto optimal (**EPO**) Mahapatra & Rajan (2020); **(5)** Continuous Pareto MTL (**CPMTL**) Ma et al. (2020). All competing methods train one model per preference vector to generate Pareto solutions. As these models optimize on each ray separately, we do not treat them as a baseline but as "*gold standard*" in term of performance. Their runtime, on the other hand, scales linearly with the number of rays.

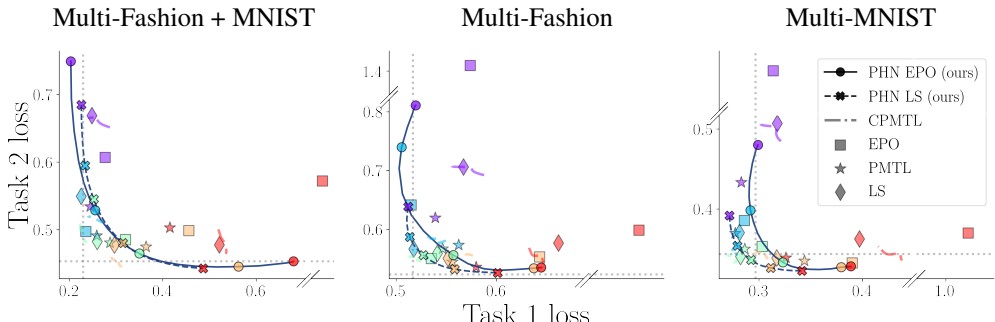

Figure 3: *Multi-MNIST*: Test losses produced by the competing method on three Multi-MNIST datasets. Colors correspond to different preference vectors, using the same color map as in Figure 1. PHN generates a continuous Pareto front (solid/dashed line for PHN-EPO/PHN-LS) while mapping any given preference to its corresponding solution (circles for PHN-EPO and x's for PHN-LS).

Genetic algorithms are perhaps the most popular approaches for MOO. Two leading approaches in that area, that also use preference information are NSGA-III (Deb et al., 2002) and MOEA/D (Zhang & Li, 2007). Unfortunately, while these methods can be applied to small networks (like the toy example in Figure 1, which gave sub-par results even after 100,000 generations), they failed to converge to meaningful solutions in our larger-scale experiments. Therefore, we do not compare PHN with GAs on large-scale setups.

**Evaluation metrics:** A common metric for evaluating sets of MOO solutions is the **Hypervolume** indicator (HV) (Zitzler & Thiele, 1999). It measures the volume in the loss space of points dominated by a solution in the evaluated set (see Appendix C). As this volume is unbounded, it is restricted to the volume in a rectangle defined by the solutions and a selected reference point. All solutions need to be bounded by the reference point, therefore we select different reference points for each experiment. In addition, we evaluate **Uniformity**, proposed by Mahapatra & Rajan (2020). This metric is designed to quantify how well the loss vector $\boldsymbol{\ell}(\theta)$ is aligned with the input ray $\boldsymbol{r}$. Formally, for any point $\theta$ in the solution space and reference $\boldsymbol{r}$ we define the non-uniformity $\mu_{\boldsymbol{r}}(\boldsymbol{\ell}(\theta)) = D_{KL}(\hat{\boldsymbol{\ell}} \| \mathbf{1}/m)$, where $\hat{\ell}_j = \frac{r_j \ell_j}{\sum_i r_i \ell_i}$. We define the uniformity as $1 - \mu_{\boldsymbol{r}}$. The main metric, HV, measures the quality of our solutions, while the uniformity measures how well the output model fits the desired preference ray. We stress that the baselines are trained and evaluated for uniformity using the same rays.

**Training Strategies:** In all experiments, our target network shares the same architecture as the baseline models. For our hypernetwork, we use an MLP with 2 hidden layers and linear heads. We set the hidden dimension to 100 for the Multi-MNIST and NYU experiment, and 25 for the Fairness and SARCOS datasets. The Dirichlet parameter $\alpha$ in Alg. 1 is set to 0.2 for all experiments, as we empirically found this value to perform well across datasets and tasks. We provide a more detailed analysis on the selection of the HN's hidden dimension and hyperparameter $\alpha$ in Appendix B.5. Extensive experimental details are provided in Appendix A.

**Hyperparameter tuning:** For PHN, we select hyperparameters based on the HV computed on a validation set. Selecting hyperparameters for the baselines is challenging because there are no clear criteria that can be computed quickly with many rays. Selecting HPs based on HV requires to train each baseline multiple times on all rays. Therefore, we select hyperparameters based on a single ray and apply the selected hyperparameters to all rays. Specifically, we collect all models trained using all hyperparameter configurations, and remove dominated solutions. Finally, we select the combination of HPs with the highest uniformity.

## 5.1 ILLUSTRATIVE EXAMPLE

We evaluate PHN on an illustrative MOO problem whose Pareto front is known (Fonseca, 1995). Here,

$$\ell_1(\theta) = 1 - \exp\left\{-\|\theta - \mathbf{1}/\sqrt{d}\|_2^2\right\}, \quad \ell_2(\theta) = 1 - \exp\left\{-\|\theta + \mathbf{1}/\sqrt{d}\|_2^2\right\},$$

Table 1: Evaluation of the different methods on three Multi-MNIST datasets.

| | Multi-Fashion+MNIST | | Multi-Fashion | | Multi-MNIST | | |
|---|---|---|---|---|---|---|---|
| | HV ⟰ | Unif. ⟰ | HV ⟰ | Unif. ⟰ | HV ⟰ | Unif. ⟰ | Run-time (min., Tesla V100) |
| LS | 2.70 | 0.849 | 2.14 | 0.835 | 2.85 | 0.846 | $9.0 \times 5 = 45$ |
| CPMTL | 2.76 | - | 2.16 | - | 2.88 | - | $10.2 \times 5 = 51$ |
| PMTL | 2.67 | 0.776 | 2.13 | 0.192 | 2.86 | 0.793 | $17.0 \times 5 = 85$ |
| EPO | 2.67 | 0.892 | 2.15 | 0.906 | 2.85 | 0.918 | $23.6 \times 5 = 118$ |
| PHN-LS (ours) | 2.75 | 0.894 | **2.19** | 0.900 | **2.90** | 0.901 | **12** |
| PHN-EPO (ours) | **2.78** | **0.952** | **2.19** | **0.921** | 2.78 | **0.920** | 27 |

with parameters $\theta \in \mathbb{R}^d$ and $d = 100$. This problem has a non-convex Pareto front in $\mathbb{R}^2$. We compared PHN with EPO, PMTL, MGDA, and two genetic algorithms (GAs) NSGA-III and MOEA/D. The results are presented in Figure 1. Linear scalarization (LS) fails to generate solutions in the concave part of the Pareto front, as theoretically shown in Boyd et al. (2004). PMTL and EPO generate solutions along the entire PF, but the solutions generated by PMTL are only optimized to be in the vicinity of the preference vector $r$. Results for MGDA and MOEA/D algorithms are shown in Appendix B.1. Our PHN-EPO generates a continuous cover of the PF in a single run. In addition, PHN-EPO returns exact Pareto optimal solutions, mapping an input preference to its corresponding point on the front. We note that similar to the EPO method, PHN-EPO can learn non-convex Pareto fronts. Additional synthetic experiments with non-convex, known Pareto fronts are presented in Appendix B.1.

## 5.2 IMAGE CLASSIFICATION

We evaluate all methods on three MOO benchmark datasets: (1) Multi-MNIST (Sabour et al., 2017); (2) Multi-Fashion, and (3) Multi-Fashion + MNIST. In each dataset, two instances are sampled uniformly at random from the MNIST (LeCun et al., 1998) or Fashion-MNIST (Xiao et al., 2017) datasets. The two images are merged into a new image by placing one at the top-left corner and the other at the bottom-right corner. The sampled images can be shifted up to four pixels in each direction. Each dataset consists of 120,000 training examples and 20,000 test set examples. We allocate 10% of each training set for constructing validation sets. We train 5 models for each baseline, using evenly spaced rays. The results are presented in Figure 3 and Table 1. The HV was calculated with reference point $(2, 2)$. PHN outperforms all other methods in terms of HV and uniformity, with significant improvement in run time. To better understand the behavior of PHN while varying the input preference vector, we visualize the per-pixel prediction contribution in Appendix B.3.

## 5.3 FAIRNESS

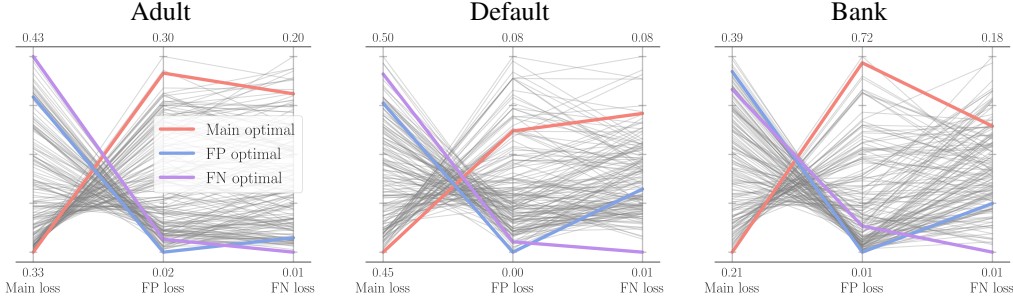

Figure 4: *Accuracy-Fairness trade-off*: The trade-off between the main classification loss and the two fairness losses, on the test sets of the three fairness datasets. Each line corresponds to a different solution along the Pareto front, generated using PHN-EPO.

Fairness has become a popular topic in machine learning in recent years (Mehrabi et al., 2019), and numerous approaches have been proposed for modeling and incorporating fairness into ML

Table 2: Comparison on three Fairness datasets. Run-time is evaluated on the Adult dataset.

| | Adult | | Default | | Bank | | |
| --- | --- | --- | --- | --- | --- | --- | --- |
| | HV ⇑ | Unif. ⇑ | HV ⇑ | Unif. ⇑ | HV ⇑ | Unif. ⇑ | Run-time (min., Tesla V100) |
| LS | 0.628 | 0.109 | 0.522 | 0.069 | 0.704 | 0.215 | 0.9 × 10 = 9.0 |
| PMTL | 0.614 | 0.458 | 0.548 | 0.471 | 0.638 | 0.497 | 2.7 × 10 = 26.6 |
| EPO | 0.608 | **0.734** | 0.537 | **0.533** | 0.713 | **0.881** | 1.6 × 10 = 15.5 |
| PHN-LS (ours) | **0.658** | 0.289 | **0.551** | 0.108 | 0.730 | 0.615 | **1.1** |
| PHN-EPO (ours) | 0.648 | 0.701 | 0.548 | 0.359 | **0.748** | 0.821 | 1.8 |

systems (Dwork et al., 2018; Kamishima et al., 2012; Zafar et al., 2019). Here, we adopt the approach of Bechavod & Ligett (2017). They proposed a 3-dimensional optimization problem, with a classification objective and two fairness objectives: False Positive (FP) fairness, and False Negative (FN) fairness. We compare PHN with EPO, PMTL, and LS on three commonly used fairness datasets: Adult (Dua & Graff, 2017), Default (Yeh & Lien, 2009) and Bank (Moro et al., 2014).

Each dataset is divided into train/validation/test sets of sizes 70%/10%/20% respectively. All baselines are trained and evaluated on 10 preference rays. We present the results in Table 2. HV calculated with reference point $(1, 1, 1)$. PHN achieves the best HV performance across all datasets, with reduced training time compared to baselines. EPO performs best in terms of uniformity, however, recall it is evaluated on its training rays. We visualize the accuracy-fairness trade-off achieved by PHN in Figure 4. We also tested GAs on this task but the runtime was extremely high (two days, compared with a runtime of minutes for gradient-based methods) with poor performance.

## 5.4 PIXEL-WISE CLASSIFICATION AND REGRESSION

Image segmentation and depth estimation are key problems in computer vision with various applications (Minaee et al., 2020; Bhoi, 2019). We compare PHN to the other methods on the NYUv2 dataset (Silberman et al., 2012). NYUv2 is a challenging indoor scene dataset that consists of 1449 RGBD images and dense per-pixel labeling. We use this dataset as a multitask learning benchmark for semantic segmentation and depth estimation tasks. We split the dataset into train/validation/test sets of sizes 70%/10%/20% respectively, and use an ENet-based architecture for the target network (Paszke et al., 2016). Each baseline method is trained five times using five evenly spaced rays. The results are presented in Table 3. HV is calculated with reference point $(3, 3)$. PHN-EPO achieves the best HV and uniformity, while also being faster than the baselines.

Table 3: Comparison on NYUv2.

| | NYUv2 | | |
| --- | --- | --- | --- |
| | HV ⇑ | Unif. ⇑ | Run-time (hours, Tesla V100) |
| LS | 3.550 | 0.666 | 0.58 × 5 = 2.92 |
| PMTL | 3.554 | 0.679 | 0.96 × 5 = 4.79 |
| CPMTL | 3.570 | - | 0.71 × 5 = 3.55 |
| EPO | 3.266 | 0.728 | 1.02 × 5 = 5.11 |
| PHN-LS (ours) | 3.546 | 0.798 | **0.67** |
| PHN-EPO (ours) | **3.589** | **0.820** | 1.04 |

Table 4: Comparison on SARCOS dataset.

| | SARCOS | | |
| --- | --- | --- | --- |
| | HV ⇑ | Unif. ⇑ | Run-time (hours, Tesla V100) |
| LS | 0.688 | -0.008 | 0.17 × 20 = 3.3 |
| EPO | 0.637 | **0.548** | 0.5 × 20 = 10 |
| PHN-LS (ours) | 0.693 | 0.020 | 0.2 |
| PHN-EPO (ours) | **0.728** | 0.258 | 0.8 |

## 5.5 MULTITASK REGRESSION

When learning with many objectives using preference-specific models, generating a good coverage of the solution space becomes too expensive in computation and storage space, as the number of rays needed grows with the number of dimensions. As a result, PHN provides a major improvement in training time and solution space coverage (measured by HV). To evaluate PHN in such settings, we use the SARCOS dataset (Vijayakumar), a commonly used dataset for multitask regression (Zhang & Yang, 2017). SARCOS presents an inverse dynamics problem with 7 regression tasks and 21 explanatory variables. We omit PMTL from this experiment because its training time is significantly

longer compared to other baselines, with relatively low performance. For LS and EPO, we train 20 preference-specific models. The results are presented in Table 4. HV calculated with reference point $(1, ..., 1)$. As before, PHN performs best in terms of HV, while reducing the overall runtime.

## 6 THE QUALITY-RUNTIME TRADE-OFF

While PHN learns the entire front in a single model, the competing methods need to train multiple models to cover the Pareto front. As a result, these methods have a clear trade-off between their performance and their run time. To better understand this trade-off, Figure 5 shows the HV and run time (in log-scale) of LS and EPO when training a various number of models. For Fashion-MNIST, we trained 25 models, and at the inference time, we selected subsets of various sizes (1 ray, 2 rays, etc) and computed their HV. The shaded area in Figure 5 reflects the variance over different selections of ray subsets. We similarly trained 30 EPO/LS models for Adult and 40 for SARCOS. PHN-EPO achieves superior or comparable hypervolume while being an order of magnitude faster. PHN-LS is even faster, but this may come at the expense of a lower hypervolume.

To further understand the source of PHN performance gain, we evaluate the HV on the same rays used for the competing methods (see Appendix B.4). In many cases, PHN outperforms models trained on specific rays, when evaluated on those exact rays. We attribute the performance gain to the inductive bias effect that PHN induces on a model by sharing weights across rays.

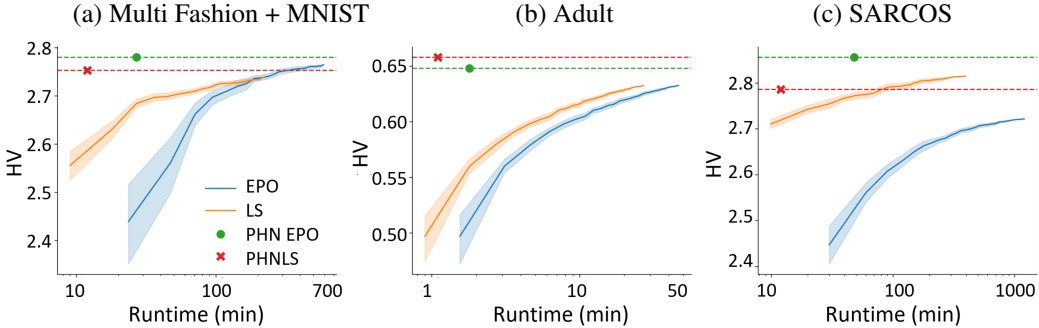

Figure 5: HV vs runtime (min.) comparing PHN with preference-specific LS and EPO. For LS and EPO, each value was computed by evaluating with a subset of models. PHN achieves higher HV significantly faster (x-axis is in log scale). Shaded area marks the variance across selected subsets.

## 7 CONCLUSION

We put forward a novel MOO setup which we term Pareto Front Learning (PFL). Learning the Pareto front using a single unified model that can be applied to a specific objective preference at inference time. We propose PHN, a model for this setup based on a hypernetwork, and evaluate it on a large and diverse set of tasks and datasets. Our experiments show significant gains in both runtime and performance on all datasets. This novel approach provides users with the flexibility of selecting the operating point at inference time and opens new possibilities where user preferences can be tuned and changed at deployment.

## ACKNOWLEDGEMENTS

This study was funded by a grant to GC from the Israel Science Foundation (ISF 737/2018), and by an equipment grant to GC and Bar-Ilan University from the Israel Science Foundation (ISF 2332/18). AS and AN were funded by a grant from the Israeli Innovation Authority, through the AVATAR consortium.

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

# Appendix: Learning the Pareto Front with Hypernetworks

## A    EXPERIMENTAL DETAILS

**Pixelwise Classification and Regression:**    For NYUv2 we performed a hyperparameters search over the number of epochs and learning rates $\{1e-3, 5e-4, 1e-4\}$. We train each method for 150 epochs with Adam (Kingma & Ba, 2015) optimizer and select an optimal set of learning rate and number of epochs according to Sec. 5. We evaluate PHN over 25 evenly spaced rays.

**Image classification:**    For the Multi-MNIST, experiments we use a LeNet-based target network (Le-Cun et al., 1998) with two task-specific heads. We train all methods using an Adam optimizer with learning rate $1e-4$ for 150 epochs and batch size 256. We then choose the best number of epochs for each method using the validation set. We evaluate PHN over 25 rays.

**Fairness:**    We use a $3-$layer feed-forward target network with hidden dimensions $[40, 20]$. We use learnable embeddings for all categorical variables. We train all methods for 35 epochs using an Adam optimizer with learning rates $\{1e-3, 5e-4, 1e-4\}$. We choose the best hyperparameter configuration using the validation set. Further details on the datasets are provided in Table 5. We evaluate PHN over a sample of 150 rays.

**Multitask regression:**    For the SARCOS experiment, we use a $4-$layer feed-forward target network with 256 hidden dimension. All categorical features are passed through embedding layers. We train all methods for 1000 epochs using an Adam optimizer and learning rates $\{1e-3, 5e-4, 1e-4\}$. We choose the best hyperparameter configuration base on the validation set. We use 40,036 training examples, 4,448 validation examples, and 4,449 test examples. We evaluate PHN over a sample of 100 rays.

Table 5: Fairness datasets.

|  | Adult | Default | Bank |
|---|---|---|---|
| Train | 32559 | 21600 | 29655 |
| Validation | 3618 | 2400 | 3295 |
| Test | 9045 | 6000 | 8238 |

## B    ADDITIONAL EXPERIMENTS

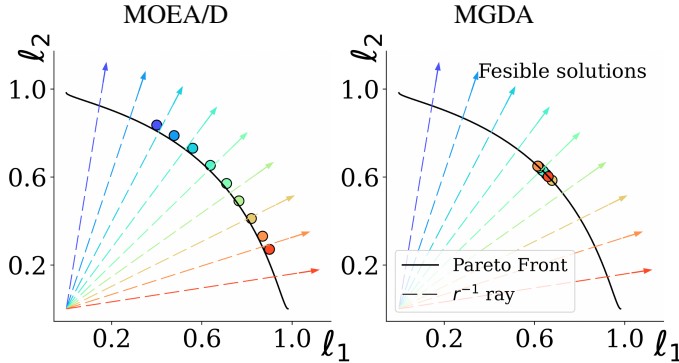

Figure 6: *Illustrative example using the popular task of Fonseca (1995)*: The genetic algorithm MOEA/D fails to reach the exact pareto front after $100,000$ generations. MGDA reaches Pareto optimal solutions, but is not suitable for generating preference-specific solutions and only covers a small fraction of the pareto front.

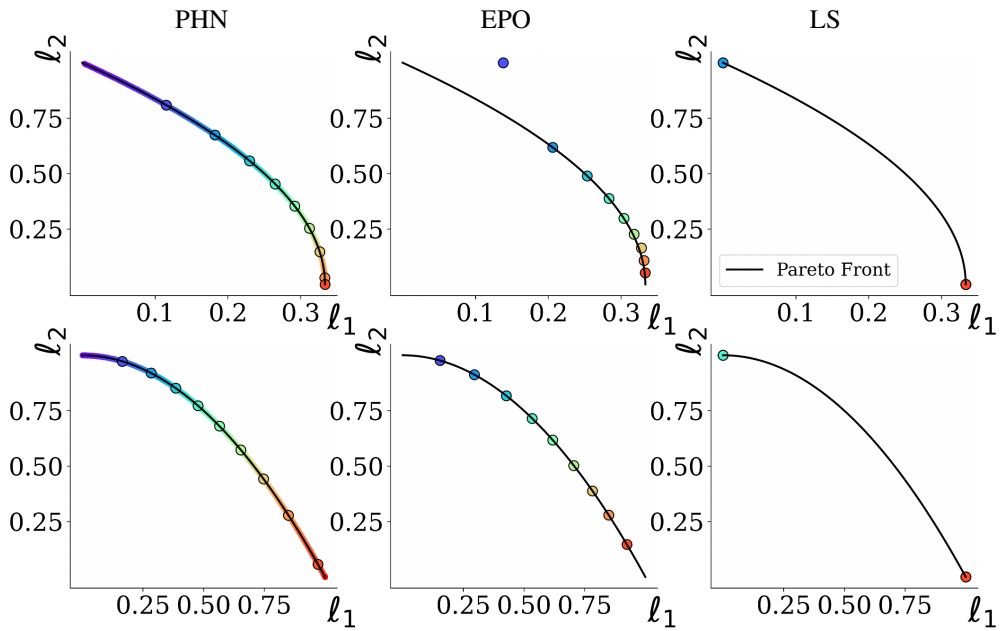

Figure 7: *Known Pareto front*: **Top:** Evaluation on Problem 2 from Evtushenko & Posypkin (2013). **Bottom:** Evaluation on ZDT2 (Zitzler et al., 2000). PHN converges to the entire PF, mapping a ray to its corresponding solution on the front. Each circle in the PHN plots denotes a solution corresponds to the baseline rays.

### B.1 KNOWN PARETO FRONT

We provide two additional experiments over popular MOO benchmarks whose Pareto front is known. In addition in Figure 6, we provide additional results for MGDA (Sener & Koltun, 2018) and MOEA/D (Zhang & Li, 2007) on the illustrative example from Section 5.1.

We evaluate PHN on two MOO problems with known, non-convex Pareto front: ZDT2 (Zitzler et al., 2000) and problem two from Evtushenko & Posypkin (2013), henceforth Problem2. In both problems, the density of solutions across the Pareto-optimal region is non-uniform.

For a solution $\theta = (\theta_1, \theta_2) \in [0,1]^2$, Problem2 is given by,

$$\ell_1(\theta) = ((\theta_1 - 1)\theta_2^2 + 1)/3, \quad \ell_2(\theta) = \theta_1,$$

and ZDT2 is given by,

$$\ell_1(\theta) = \theta_1, \quad \ell_2(\theta) = 1 - \left(\frac{\theta_1}{1 + 9\theta_2}\right)^2.$$

The results are presented in Figure 7. PHN-EPO converges to the entire PF using a single HN model. in contrast with PHN-EPO, EPO fails to converge to the PF for all rays in ZDT2.

### B.2 GENERALIZATION TO UNSEEN RAYS

When the number of objectives is large, the curse of dimensionality makes it likely that rays encountered in test time are "far" from rays encountered during training. Although we evaluated PHN on unseen rays[1] through all experiments, here we would like to better examine the generalization of PHN in that scenario. We conducted controlled experiment on SARCOS dataset with 7 tasks. During training, we sample rays from a predefined grid on the $7-$dimensional simplex. We use the structured approach of Das & Dennis (1998) for generating well-spaced points. During inference, we measure HV and Uniformity on two sets of rays: (i) Randomly sampled train rays (ii) Randomly sampled test rays from a shifted grid, so they are most distant from the train grid. In both cases, test samples differ

---

[1]With $\sim$one probability.

from training samples. We find that testing with unseen test rays only causes a minor decrease in both HV and Uniformity compared with train rays. The HV is reduced from $0.7337$ to $0.7336$ and uniformity is reduced from $0.163$ to $0.158$. This experiment indicates that PHN generalizes well to unseen rays, even in problems with many objectives. We note that the HV results presented here are higher than the ones reported in Section 5.5. We attribute this inconsistency to change of preference vectors distribution used during training. While throught the paper we sample preference rays from a Dirichlet distribution with $\alpha = 0.2$, here we sample from a fixed, evenly spaced grid. We provide analysis on the robustness to the choice of the Dirichlet distribution parameter $\alpha$ in Appendix B.5.

## B.3 MNIST PHN INTERPRETATION

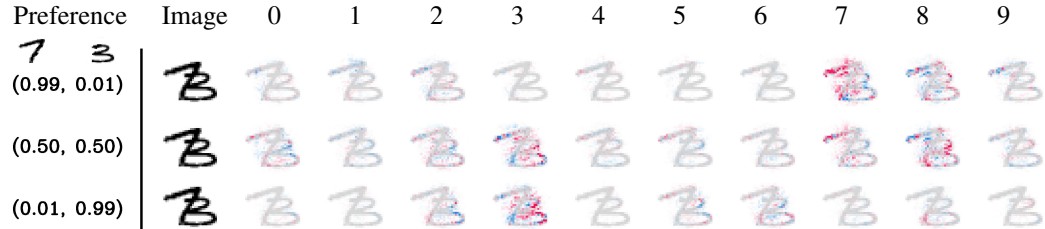

Figure 8: *PHN interpretation*: PHN is provided with a sample from the Multi-MNIST dataset, where an image of 3 and an image 7 are overlayed. We show the effect that conditioning on different preference rays (*left*) has on the decision made by the model. Red pixels denote increased contribution to the prediction. See text for details.

To better understand the behaviour of PHN while varying the input preference vector, Figure 8 shows the per-pixel contribution to prediction. We use the SHAP framework (Lundberg & Lee, 2017) for visualization. Red pixels denote an increased probability for a given class, while blue pixels denote a decreased probability. The top left and bottom right digits are labeled as $\{7, 3\}$ respectively. In the first row (ray: $(0.99, 0.01)$), the model is focused on task 1 (with label 7), most of the red pixels come from the top left digit in column 7, as expected. Similarly, in the third row (ray: $(0.01, 0.99)$) most of the red pixels are focused on the bottom right digit in column 3.

## B.4 SAME-RAY EVALUATION

In Section 5 of the main text we evaluate PHN on multiple evenly spaced rays. For completeness, we report here the HV and Uniformity achieved by PHN, while evaluated on the same rays used for training and evaluating the baseline methods. The results are provided in Table 6. Surprisingly, in many cases PHN achieves better performance. We attribute this to the inductive bias effect of sharing weights across all preference rays.

Table 6: Evaluation of PHN on baselines' ray.

|  | Fashion+MNIST | | Multi-Fashion | | Multi-MNIST | | Adult | | Default | | Bank | | NYUv2 | | SARCOS | |
|---|---|---|---|---|---|---|---|---|---|---|---|---|---|---|---|---|
|  | HV | Unif. | HV | Unif. | HV | Unif. | HV | Unif. | HV | Unif. | HV | Unif. | HV | Unif. | HV | Unif. |
| PHN-LS | 2.740 | 0.852 | 2.190 | 0.838 | 2.900 | 0.840 | 0.624 | 0.282 | 0.538 | 0.106 | 0.695 | 0.603 | 3.494 | 0.683 | 0.693 | -0.051 |
| PHN-EPO | 2.760 | 0.908 | 2.180 | 0.860 | 2.760 | 0.857 | 0.631 | 0.742 | 0.531 | 0.395 | 0.717 | 0.833 | 3.565 | 0.701 | 0.728 | 0.190 |

## B.5 DESIGN CHOICE ANALYSIS

In Section 5 we used fixed values of the hidden dimension and the Dirichlet parameter $\alpha$ (see Alg. 1). Here we examine the effect of different choices for these hyperparameters. We use the Multi-Fashion + MNIST dataset, and the PHN-EPO variant of our method. We first fix the hidden dimension to 100, and vary $\alpha$ from $0.1$ to $1$ (uniform sampling). The results are presented in Figure 9(a). It appears that at least for this dataset, PHN is quite robust to changes in $\alpha$. Next, we set $\alpha = 0.2$ and examine the effect of changing the dimension of the hidden layers in our hypernetwork. We present the results in Figure 9(b). Here, the effect on the HV is stronger, with best HV achieved using hidden dimension of 100.

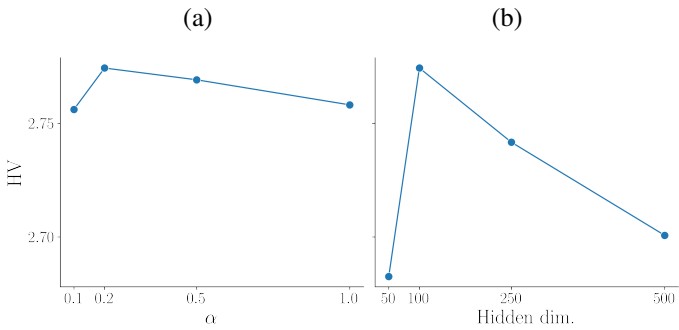

Figure 9: *Hyperparameters analysis*: Performance analysis on the Multi-Fashion + MNIST dataset, for different architecture designs and sampling choices.

## B.6 ADDITIONAL RUNTIME ANALYSIS

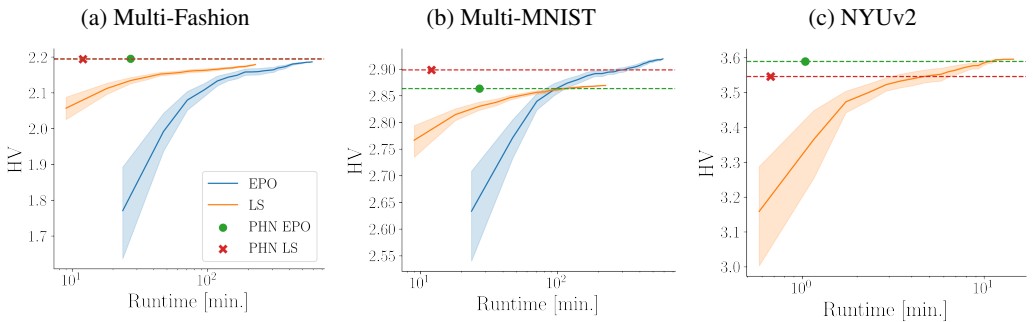

Figure 10: Runtime analysis for NYUv2, and two additional Multi-MNIST dataset.

We extend the runtime analysis from Section 5.5. Figure 10 provides additional runtime analysis on the Multi-MNIST, Multi-Fashion, and NYUv2 datasets. For NYUv2 we omit the EPO method, since it was outperformed by LS in both HV and runtime.

## C HYPERVOLUME

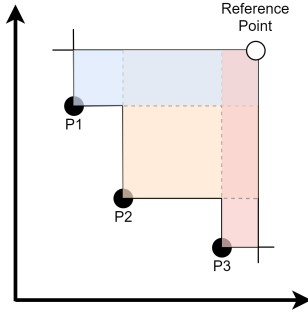

Figure 11: *Hypervolume illustration*: The colored area represents the hypervolume obtained by Pareto set $\{P_1, P_2, P_3\}$ with respect to the reference point.

The hypervolume metric is defined as follows: Given a set of points $S \subset \mathbb{R}^n$ and a reference point $\rho \in \mathbb{R}^n_+$, the hypervolume of $S$ is measured by the region of non-dominated points bounded above by $\rho$.

$$HV(S) = VOL\big(\{q \in \mathbb{R}^n_+ \mid \exists p \in S : (p \preceq q) \wedge (q \preceq \rho)\}\big),$$

where $VOL$ is the euclidean volume. We can interpret hypervolume as the measure of the union of the boxes created by the non-dominated points.

$$HV(S) = VOL \left( \bigcup_{\substack{p \in S \\ p \preceq \boldsymbol{\rho}}} \prod_{i=1}^{n} [p_i, \rho_i] \right)$$

As a measure of quality for MOO solutions, it takes into account both the quality of individual solutions, which is the volume they dominate, and also the diversity, measured in the overlap between dominated regions. See example in Fig. 11.

