# OpenReview forum: "Learning the Pareto Front with Hypernetworks"
_ICLR.cc/2021/Conference — ICLR 2021 Poster_

### Official Review · AnonReviewer1 · 2020-10-26
**Learning the Pareto Front with Hypernetworks**

**Rating:** 6
**Confidence:** 4

**Review:**

This paper tracks the problem of learning the entire Pareto front to allow the user to select a desired Pareto optimal solution by one inference procedure without retraining the model. The high-level idea is to learn the entire Pareto front simultaneously using a single hyper network, which receives as input the desired preference vector and returns a Pareto-optimal solution whose loss vector is in the desired direction. The paper gives an early trial to build a toolbox to allow users to get a desired solution by a single inference procedure.

Strength:
1.	This paper is an early trial to use a hyper network to directly approximate the Pareto Optimal front, allowing practitioners to flexibly choose Pareto solutions conditioned on different preference vectors.
2.	The paper is well-written and a substantial number of experiments are conducted, promising good results of the proposed method.

Feedbacks:
1.	My major concern is that there is a similar work [1] that shares the same spirit with the work.  The authors might want to clarify the differences or conduct some experimental comparisons.
2.	It is better to attach more details about the EPO algorithm.
3.	There are some typos or grammar issues, such as 1) Page 6, the title of the subsection ‘Hyperparamter tuning’ should be ‘Hyperparameter tuning’. 2) Page 6, ‘We therefor’ should be ‘We therefore’. 3) Page 6, ‘as follow’ should be ‘as follows’. 4) Page 7, ‘are train and evaluate’ should be ‘are trained and evaluated’.

[1] X. Lin, Z. Yang, Q. Zhang, and S. Kwong, “Controllable Pareto multi-task learning,” arXiv preprint arXiv: 2010.06313, 2020.

---

> ### Author Response · Authors · 2020-11-15
> **Response to AnonReviewer1**
>
> Thank you for your helpful and thoughtful feedback. We provide our response below.
>
> **Recent similar work:**
>
> Thank you for pointing out this highly relevant work. We stress that it is a concurrent work that was published on arxiv after our own work was submitted to ICLR. It was also submitted to ICLR 2021 (https://openreview.net/forum?id=5mhViEOQxaV). We find it encouraging that other researchers find the problem addressed in our paper, namely approximating the entire Pareto front using a single model, interesting and relevant. We revise our paper to acknowledge this work.
>
> **Elaborate on EPO:**
>
> We added more details on the EPO approach [Section 2] and how we use it in our proposed method PHN-EPO [Section 4].
>
> **Typos:**
>
> Thank you, we fixed all typos and grammar issues.

---

### Official Review · AnonReviewer2 · 2020-10-28
**Good paper with minor issues**

**Rating:** 7
**Confidence:** 4

**Review:**

The paper proposed a novel algorithm for MOO, which combines preference-based methods with hypernetworks, in order to encapsulate the preferences in the network input.
The paper is well written, the proposed method is clear, and the experiments are sufficient.
My concerns are the following.

First, in the abstract you say that
"Recent optimization algorithms can target a specific desired ray in loss space,
but still face two grave limitations: (i) A separate model has to be learned for
each point on the front; and (ii) The exact trade-off must be known prior to the
optimization process. "

This is not entirely true. In Reinforcement Learning (RL) manifold-based approach such as

Parisi et al, "Multi-objective Reinforcement Learning through Continuous Pareto Manifold Approximation"
Yang et al, "A Generalized Algorithm for Multi-Objective Reinforcement Learning and Policy Adaptation"

do not need neither a separate model, nor to know the trade-off a priori. The work of Parisi et al learns a parametrization producing infinitely many solutions at the same time using a specific loss, while Yang et al train a neural network which takes preferences over the objectives as input, and generalizes over them.
Despite being tested only on MORL, these algorithms can easily be extended to MOO.
Therefore, I suggest you to rephrase your abstract and introduction to mention these methods.

In particular, the work of Yang et al is very reminiscent of your algorithm, since both include the preference vector in the network input.

Nonetheless, I feel the paper has enough novely thanks to the use of hypernetworks, EPO, and its application to MOO.
The evaluation is sufficient, but I would suggest to move the evaluation of evolutionary algorithms to the main section. Evolutionary algorithms are extremely popular in MOO and thanks to a wide variety of fitness functions can learn any frontier, even though they may require longer run time (as your experiments clearly show).

Finally, a bit more analysis on concave frontier would be beneficial. In Figure 1 you show that LS fails in finding concave frontiers, but it seems to me that all experiments have convex frontier. To my understanding, EPO should address this limitation ("The recent EPO algorithm can guarantee convergence to a local Pareto optimal point on a specified ray r, addressing the theoretical limitations of LS"). Am I correct? A bit more discussion and experiments with concave frontiers would be a nice addition to the paper.

Overall, I am leaning to accept it but the authors should address the above issues.

**EDIT **
The authors have addressed my concerns, and I have increased my score.

---

> ### Author Response · Authors · 2020-11-15
> **Response to AnonReviewer2**
>
> Thank you for your helpful and thoughtful feedback. We provide our response below.
>
> **Additional relevant work:**
>
> Thank you for bringing this to our attention. We acknowledged these relevant studies in the revised paper and rephrased the abstract and introduction accordingly.
>
> **Move EA evaluation to the main text:**
>
> Following your suggestion, we moved the evaluation of the genetic algorithm NSGA-III to the main text (Figure 1). We also discuss genetic algorithms in more detail in Sec. 5.1.
>
> **A bit more discussion and experiments with concave frontiers would be a nice addition to the paper:**
>
> EPO (as well as the other MOO approaches) can converge to the non-convex parts of the Pareto front. As a result, PHN-EPO can also learn non-convex frontiers.
> Following this comment, we made the following changes to the paper: First, figure 1 corresponds to a popular MOO benchmark [1] with a concave frontier. We now provide the details for Figure 1 in the main paper [Section 5.1]. Second, we also add two new experiments on two MOO benchmarks [2,3] that have known, non-convex Pareto fronts. The results are qualitatively similar to Figure 1, and are provided in Appendix C.1, Figure 7. The evaluations demonstrate that PHN-EPO can generate preference-specific solutions along the entire Pareto front, even for non-convex problems.
>
> **Citations:**
>
> [1] Carlos Manuel Mira da Fonseca.Multiobjective genetic algorithms with application to control engineering problems. PhD thesis, University of Sheffield, 1995.
>
> [2] Yu G Evtushenko and Mikhail Anatol’evich Posypkin. Nonuniform covering method as applied to multicriteria optimization problems with guaranteed accuracy.Computational Mathematics and Mathematical Physics, 53(2):144–157, 2013.
>
> [3] Eckart Zitzler, Kalyanmoy Deb, and Lothar Thiele.  Comparison of multiobjective evolutionary algorithms: Empirical results.Evolutionary computation, 8(2):173–195, 2000.

---

### Official Review · AnonReviewer3 · 2020-10-28
**The idea of learning the entire Pareto front is interesting. However, the organization and experiments need more elaboration.**

**Rating:** 6
**Confidence:** 3

**Review:**

Thanks for the efforts of the authors. After reading the response and other reviews,  I raise my score from 5 to 6. However, the proposed algorithm lacks analysis. So I cannot improve my score further.
___________________________________________________

Summary:
The paper proposes using a hypernetwork to learn the entire Pareto front of a multi-objective optimization.  It develops two approaches using linear scalarization and exact Pareto optimal (EPO), respectively. Experiments on several multi-task learning tasks show its advantage in terms of hypervolume and uniformity.

Strengths:
+ It is very interesting and useful that the paper tries to learn the entire Pareto front directly.
+ The paper proposes using hypernetworks to learn the Pareto front for multi-task learning.

Weaknesses:
- Algorithm 1 is the main contribution. The pseudocode and its descriptions are not clear. For example, it is better to use two algorithms to descript PHN-LS and PHN-EPO, respectively. r is the weights for LS, while it is the ray for EPO. However, r should be pre-given. How and why PHN could optimize the multi-objective optimization for any r? What does “Dir(\alpha)” mean?
-PHN is a solving method for MOO problems. So the paper should verify that it could learn the entire Pareto front. The paper may test with MOO problems with known Pareto fronts. The results in Figure 2 cannot testify to this claim. This is more important than the experiments on the multi-task learning.

Minor comments:
1.	The legend in Figure2 should be colored.
2.	The captions of the subfigures in Figure 4 should be under them.
3.	Typos:
sampled “form” the m-dimensional-> sampled “from” the m-dimensional, “minimas”->”minima”, longer “then”-> longer “than”, “follwoing" ->“following"

---

> ### Author Response · Authors · 2020-11-15
> **Response to AnonReviewer3**
>
> Thank you for your helpful and thoughtful feedback. We provide our response below.
>
> **Clarify inference and training procedures:**
>
> Following your comment we revised the text [Section 4] to better explain our training and inference procedures: During training, we sample $r$ at each iteration from the Dirichlet distribution with parameter $\alpha$ ($Dir(\alpha)$). During inference, the user is free to choose a preferred operating point (preference vector $r$) that is simply given as input to the hypernetwork. The PHN outputs model weights tuned for that preference vector.
>
> **Add experiments with known Pareto front:**
>
> Following your comments, we made the following changes: First, we added two new experiments on two popular MOO benchmarks [2,3] which have a known, non-convex Pareto front [Appendix C.1, Figure 7]. The evaluation shows that PHN can learn the entire Pareto front in these problems.
>
> Second, we added a section (5.1) that discusses in more detail the illustrative example of Figure 1. This problem is a well-known MOO benchmark with a non-convex front [1].
>
> Finally, we point out that experiments on MTL are important as they show that the method works on real challenging problems.
>
> **Graphics:**
>
> Clarification of figure 3 (revised manuscript) - the colors represent different preference vectors $r$ in the objective space. The legend is used to distinguish between the different methods.
>
> **Typos:**
>
> Thank you, we fixed all typos and grammar issues.
>
> **Citations:**
>
> [1] Carlos Manuel Mira da Fonseca.Multiobjective genetic algorithms with application to control engineering problems. PhD thesis, University of Sheffield, 1995.
>
> [2] Yu G Evtushenko and Mikhail Anatol’evich Posypkin. Nonuniform covering method as applied to multicriteria optimization problems with guaranteed accuracy. Computational Mathematics and Mathematical Physics, 53(2):144–157, 2013.
>
> [3] Eckart Zitzler, Kalyanmoy Deb, and Lothar Thiele. Comparison of multiobjective evolutionary algorithms: Empirical results. Evolutionary computation, 8(2):173–195, 2000.

---

### Official Review · AnonReviewer4 · 2020-10-28
**Learning the Pareto Front**

**Rating:** 6
**Confidence:** 3

**Review:**

The paper proposes a method for multi-objective optimization. The key idea is to learn the entire Pareto front at once by training a hypernetwork that takes preference vector as an inputs and outputs network parameters, which corresponds to a point on the Pareto set with the desired trade-off specified by the preference vector. Specifically, the hypernetwork is a multi-head network where each head outputs a weight tensor of a module in the target network. The method improves HV from the baselines, in several multi-task learning problems, including image classification, regression and, mixed classification and regression.

+) The main contribution of this work is to learn a continuous function that maps a preference vector to network parameters that corresponds to the desired trade-off. The trained hypernetwork generalizes to preference vectors unseen during training so that the required training time for getting models of an arbitrary trade-off is reduced.

+) Compared to prior work, CPMTL, which starts from a Pareto optimal point and extends Pareto front locally around the point, the proposed method attempts to train a single hypernetwork that represents the entire Pareto front.

+) Pareto front provides insights on the trade-off relation between tasks but it usually requires repetitive training of models under different preference settings. I think this method can be useful when understanding the relations among tasks by significantly reducing the training time to obtain the Pareto front.

-) One concern is the scalability to the number of tasks. The amount of reduce in the training time of the proposed method over the base methods depends on how much the method generalize to the preference vectors unseen during training. For example, ideally, the model trained using the preference vectors [0,1] and [1,0] generalizes to [0.5,0.5]. In other words, it is desired that a hypernetwork is trained using less number of samples while generalizes well to arbitrary trade-offs, relying on the smoothness of the hypernetwork. Otherwise, the training requires more number of iterations to sample sufficient number of preference vectors that covers the whole preference vector space to match the original performance, where its size grows exponentially to the number of tasks. Current manuscript lacks experimental or theoretical analysis on this generalization performance.

-) Relating to the above point, I think evaluation measure HV and uniformity are not sufficient to evaluate the overall behavior. HV measures the quality of exploration, but higher HV does not necessarily mean that every model of method A dominates method B. For example, in the left figure of Figure 2, some points of EPO dominates PHN-EPO though HV is higher for PHN-EPO (Table 1). This implies that the proposed method reduces training time at cost of possible degrading movement of Pareto front. In other words, depending on the hyperparameter alpha used for the Dirichlet distribution, under the comparable training time to the baseline, the baseline method may dominate its PHN counter part at certain preferences.

-) And therefore, I think one needs to investigate the accuracy plot (as Figure 2) together with HV and uniformity to better understand the behavior of the method. Some are missing in the current manuscript (e.g. NYUv2)

-) Another concern is the scalability to the target model capacity. In the current manuscript, all the experiments are performed using architectures with small number of parameters (up to \~0.37M of ENNet), which is much smaller than popular architectures such as ResNet50 (\~26M). In addition, the required size of the hypernetwork grows at least proportional to the target model capacity,
and the amount of training data and the training time likely increase accordingly.

-) HyperNetwork require some overhead of memory and computation cost during the test time.

-) Comparison with important prior work CPMTL is necessary.

I would recommend 'accept'. Though I have concerns about the scalability and lack of analysis, I think this work has some advantages stated above.

Questions:
1. Discussion on the generalization performance to unseen preference vectors
2. Comparison with CPMTL
3. Accuracy plot for NYUv2

___
Thanks for the response. After reading the authors' response and other reviews, I would like to keep my recommendation.

---

> ### Author Response · Authors · 2020-11-15
> **Response to AnonReviewer4 - Part 1**
>
> Thank you for your helpful and thoughtful feedback. We provide our response below.
>
> **Scalability (to tasks) and generalization:**
>
> Thank you for the insightful observation. First, we stress that the experiments show that PHN generalizes well, because we test it on unseen preference directions. Following your suggestion, and with the purpose of quantifying generalization to rays that were not seen during training, we conducted an additional, controlled experiment on SARCOS dataset with 7 tasks. During training, we sample rays from a predefined grid. During inference, we measure HV and Uniformity on two sets of rays: (i) Randomly sampled train rays (ii) Randomly sampled test rays from a shifted grid, so they are most distant from the train grid. In both cases, test *samples* are different from training samples.  We find that testing with unseen test rays only causes a minor decrease in both HV and Uniformity compared with train rays. This is consistent with the reviewer's suggestion that the smoothness of the PHN allows it to generalize to unseen rays. Full details of this new experiment are now given in Appendix C.2.
>
> In addition, we point out that most MOO and MTL benchmarks contain less than 7 tasks. Our SARCOS experiment already considered a setting with higher dimensionality (in terms of num. objectives) than typical MOOs.
>
> **Hypervolume as an evaluation measure:**
>
> This comment raises a deep question about the right way to **evaluate a set of solutions**. HV is the standard, and most commonly used metric for evaluating a set of solutions, as required in MOO settings [1,2,3]. As such, and to be consistent with the literature, we adopted HV as our main metric through all experiments. Importantly, HV measures more than just exploration, but also the quality of the results, similar to AUC. Indeed, for some preferences, a baseline model might dominate PHN. However, the higher HV obtained by PHN suggests that, on average, PHN dominates the baselines. In Appendix C.4 we show that even when evaluated on the rays that are used to train the baselines, PHN can outperform all baselines (e.g., for all multi-MNIST experiments). This is a somewhat surprising results, as the baselines train a separate model for each ray, and is being tested on that same ray. We attributed this to the inductive bias effect of sharing weights across all preference rays. Lastly, we point out that the Dirichlet parameter alpha is only used at training time by PHN, and is not being used for inference or training. All baselines are always trained and tested on the same evenly spaced rays.
>
> **Alternative performance metrics, report accuracy on NYUv2:**
>
> We adopted HV as our main metric to align with the MOO literature. There are several issues with using accuracy as a quality measure, which caused the community to avoid it, and prevents us from using it in this paper. (1) First, there is no standard way to evaluate accuracy on a set of solutions, as required in MOO problems, because the accuracy of different tasks should be aggregated, and there is no agreement about the weighted aggregation procedure. (2) Second, unlike HV, accuracy is only applicable to classification tasks. Specifically, it cannot be used to evaluate NYUv2 because it contains a regression task.
>
> **PHN scalability:**
>
> Indeed, the HN size grows linearly with the target network size. However, there are several simple extensions that can be applied to PHN, and make it scale to much larger networks: (i) First, one can learn only part of the target network parameters using HN. For instance, lower layers in vision applications tend to be highly conserved across classes and often even across tasks [4]. (ii) One can use different parts of the ray embedding to generate different weight tensors of the target network, mainly, provide each head of the HN with only a small part of the shared representation; or reduce the embedding size as shown in the ablation study in Appendix C.5.
>
> Following your comment, we added a discussion on the scalability of our method in Appendix B.

---

> > ### Author Response · Authors · 2020-11-15
> > **Response to AnonReviewer4 - Part 2**
> >
> > **Memory and computation cost during inference:**
> >
> > In the standard setting, in which a decision-maker (DM) selects one operating point, there is no overhead at test time. In that case, the HN is used once to generate the desired operating point (target model weights) and only this model is used later on, with no overhead. If the DM wants **flexibility at real-time**, it comes with the cost of memory and computational overhead.
> >
> > **Comparison with CPMTL:**
> >
> > In our original submission, we compared with CPMTL on all multi-MNIST experiments, and found that our approach PHN outperforms it in terms of HV [Table 1]. Following this comment, in the revised paper we add an additional comparison to CPMTL on the NYUv2 experiment [please see Table 3]. The evaluation shows that our approach provides better coverage of the solution space and trade-off curve (measured by HV). We note that CPMTL is evaluated on 55 models (5 initial solutions + 10 extended models per initial solution).
> >
> > **Citations:**
> >
> > [1] Guerreiro, Andreia P., Carlos M. Fonseca, and Luís Paquete. "The Hypervolume Indicator: Problems and Algorithms." arXiv preprint arXiv:2005.00515 (2020).
> >
> > [2] Deb, Kalyanmoy. Multi-objective optimization using evolutionary algorithms. Vol. 16. John Wiley & Sons, 2001.
> >
> > [3] Zitzler, Eckart, Dimo Brockhoff, and Lothar Thiele. "The hypervolume indicator revisited: On the design of Pareto-compliant indicators via weighted integration." In International Conference on Evolutionary Multi-Criterion Optimization, pp. 862-876. Springer, Berlin, Heidelberg, 2007.
> >
> > [4] Luan, Shangzhen, Chen Chen, Baochang Zhang, Jungong Han, and Jianzhuang Liu. "Gabor convolutional networks." IEEE Transactions on Image Processing 27, no. 9 (2018): 4357-4366.

---

### Author Response · Authors · 2020-11-15
**To All Reviewers**

We thank the reviewers for their helpful and thoughtful feedback. We are encouraged that the reviewers find our work interesting and useful (R3, R4) with enough novelty (R2)  and the paper well-written (R1, R2). They also found the approach to significantly reduce the training time (R4), the experiments substantial (R1, R2), and the results promising (R1) and improve over baselines in several MTL problems (R4).

We revised the manuscript accordingly to resolve the reviewers’ concerns, and add additional experiments and discussions to address the reviewers’ suggestion. We provide a detailed description of the changes we made in our responses below.

---

### Decision · Program_Chairs · 2021-01-07
**Final Decision**

**Decision:**

Accept (Poster)

**Comment:**

The paper proposes a hyper-net method for multi-objective optimization, which trains a neural network that maps preference vector to the corresponding Pareto solution. The proposed idea is interesting and useful, although the evaluation of the work is not overwhelming convincing. The writing of the work can be further improved.

Also, the basic idea of the work is the almost the same as a concurrent work "Lin et al 2020. controllable pareto multi-task learning" which is also submitted to this conference. The paper cited that paper briefly, "... The proposed method is conceptually similar to our approach...",  which is too vague and brief. We urge the author to provide a through discussion on the detailed difference and similarity of the works, including empirical comparisons when necessary.